# Crystal Structure of a Retroviral Polyprotein: Prototype Foamy Virus Protease-Reverse Transcriptase (PR-RT)

**DOI:** 10.3390/v13081495

**Published:** 2021-07-29

**Authors:** Jerry Joe E. K. Harrison, Steve Tuske, Kalyan Das, Francesc X. Ruiz, Joseph D. Bauman, Paul L. Boyer, Jeffrey J. DeStefano, Stephen H. Hughes, Eddy Arnold

**Affiliations:** 1Center for Advanced Biotechnology and Medicine (CABM), Rutgers University, Piscataway, NJ 08854, USA; jjharrison@ug.edu.gh (J.J.E.K.H.); s_tuske@hotmail.com (S.T.); kalyan.das@kuleuven.be (K.D.); xavier@cabm.rutgers.edu (F.X.R.); marinesci@me.com (J.D.B.); 2Department of Medicinal Chemistry, Ernest Mario School of Pharmacy, Rutgers University, Piscataway, NJ 08854, USA; 3Department of Chemistry, University of Ghana, Legon P.O. Box LG 56, Ghana; 4Department of Microbiology, Immunology and Transplantation, Rega Institute, KU Leuven, 3000 Leuven, Belgium; 5HIV Dynamics and Replication Program, National Cancer Institute, Frederick, MD 21702, USA; boyerp@mail.nih.gov (P.L.B.); hughesst@mail.nih.gov (S.H.H.); 6Department of Cell Biology and Molecular Genetics, University of Maryland College Park, College Park, MD 20742, USA; jdestefa@umd.edu; 7Department of Chemistry and Chemical Biology, Rutgers University, Piscataway, NJ 08854, USA

**Keywords:** PFV, Pol polyprotein, PR-RT, protease, reverse transcriptase, maturation

## Abstract

In most cases, proteolytic processing of the retroviral Pol portion of the Gag-Pol polyprotein precursor produces protease (PR), reverse transcriptase (RT), and integrase (IN). However, foamy viruses (FVs) express Pol separately from Gag and, when Pol is processed, only the IN domain is released. Here, we report a 2.9 Å resolution crystal structure of the mature PR-RT from prototype FV (PFV) that can carry out both proteolytic processing and reverse transcription but is in a configuration not competent for proteolytic or polymerase activity. PFV PR-RT is monomeric and the architecture of PFV PR is similar to one of the subunits of HIV-1 PR, which is a dimer. There is a C-terminal extension of PFV PR (101-145) that consists of two helices which are adjacent to the base of the RT palm subdomain, and anchors PR to RT. The polymerase domain of PFV RT consists of fingers, palm, thumb, and connection subdomains whose spatial arrangements are similar to the p51 subunit of HIV-1 RT. The RNase H and polymerase domains of PFV RT are connected by flexible linkers. Significant spatial and conformational (sub)domain rearrangements are therefore required for nucleic acid binding. The structure of PFV PR-RT provides insights into the conformational maturation of retroviral Pol polyproteins.

## 1. Introduction

Foamy viruses (FVs), the most ancient retroviruses, have the same genome organization as retrotransposons and the architecture of their structural and enzymatic proteins is also similar [1,2,3,4]. FVs are known to cause persistent infections in humans and other vertebrate hosts but, remarkably, are apathogenic in humans [5]. Many RNA viruses, retroviruses, and retrotransposons synthesize polyproteins that are proteolytically processed by cognate viral protease(s) into their functional mature forms [6,7,8,9]. Of the proteins encoded in the retroviral genome, reverse transcriptases (RTs) and proteases (PRs) are the most conserved structurally and functionally [10,11,12], despite having low sequence identity. Lentiviruses, such as HIV-1, synthesize their enzymatic proteins as components of Gag-Pol polyprotein fusions. The Pol portion of Gag-Pol polyproteins is processed into the mature enzymes (PR, RT, and IN). The Gag-Pol polyprotein is synthesized from the same full-length RNA that is used to make Gag by a −1 translational frameshift that happens approximately 5% of the time the RNA is translated. However, spuma retroviruses, such as the prototype foamy virus (PFV), synthesize their Pol polyproteins from an mRNA that is spliced from genomic RNA [13,14]. FV polyproteins undergo only minimal proteolytic processing during virion maturation. The only processing of the Gag polyprotein is a cleavage that releases a 3 kDa C-terminal peptide [9]. A cleavage between the N-terminus of integrase (IN) and the C-terminus of the RNase H domain of RT produces mature IN and PR-RT from the Pol polyprotein. The mature PR-RT has all of the enzymatic functions of PR and RT [reviewed in [15]].

The limited proteolysis of FV Gag results in infectious FV particles that are morphologically similar to immature lentiviral capsids, although this minimal processing is absolutely required for infectivity [2,9]. The PR-RTs of FVs are monomeric in solution and the RT component is responsible for converting the (+) FV ssRNA genome into dsDNA for integration into the host chromosome. In contrast to most retroviruses, in which reverse transcription takes place in newly infected cells, in FVs, reverse transcription occurs primarily in producer cells, resulting in infectious viral particles that contain dsDNA [2,4,16].

The aspartyl PR of all retroviruses, including FVs, is responsible for the proteolytic processing of polyproteins and, to be enzymatically active, retroviral PRs must form a homodimer [17]. Hence, the PR component of either PR-RT or PR-RT-IN must dimerize for the protease to be active. The mechanism(s) by which this is orchestrated remain(s) under investigation, potentially involving viral RNA or the IN domain of the Pol [15,18,19]. There are numerous structures of retroviral PRs both unliganded and in complexes with relevant substrates and inhibitors, which help to explain the mechanisms of catalysis, inhibition, and drug resistance. Similarly, there are a number of retroviral RT structures, both unliganded and in complexes with nucleic acid substrates, and inhibitors that have helped to elucidate the mechanisms of catalysis, inhibition, and drug resistance. However, there is little structural information on the structures of the Pol and Gag-Pol polyproteins: we lack an understanding of (i) how retroviral PRs dimerize in the context of the Pol polyprotein; and (ii) how the initial proteolytic processing steps are carried out when PR is still embedded in the polyprotein precursor.

How the asymmetric conformation of the subunits of the heterodimeric HIV-1 RT arises from monomeric precursors, producing the p66 (catalytic) and p51 (scaffolding) subunits, remains an unsolved puzzle. However, there are suggestions (albeit with limited structural evidence) that monomeric HIV-1 p66 can adopt a more thermodynamically favorable p51-like fold in vitro, occasionally sampling the open conformation seen in HIV-1 p66 subunit in the heterodimer, prior to dimerization and subsequent maturation [20,21,22]. However, the relevance of the formation of p66/p66 homodimer to the viral life cycle has not been demonstrated in vivo. In an effort to further investigate the structure and maturation of retroviral Gag-Pol and Pol polyproteins, we have determined and report here the crystal structure of the wild-type full-length PFV PR-RT protein at 2.9 Å resolution. The PR-RT we crystallized has a mutation in the active site of PR. Although the protein we crystalized has both polymerase and RNase H activity, the structure is in a configuration that could have neither protease nor polymerase activity. However, the solved structure is a precursor to a configuration that is competent for these activities.

The arrangement of the subdomains of PFV RT is similar to that of the p51 subunit of HIV-1 RT while the PR structurally resembles a single subunit of the catalytically active HIV-1 PR dimer. The structural organization of PFV PR-RT offers insights into the potential mechanism of dimerization of PR in the context of a polyprotein, offering a glimpse of the initial events leading up to the proteolytic processing of Pol polyproteins. The structure reveals new interfaces that are not present in the individual enzymes and provides support for the concept of using structural data to develop small molecules that target the immature enzymes of retroviruses.

## 2. Materials and Methods

### 2.1. Cloning, Expression and Purification

PFV PR-RT, with a protease null mutation (D24A), WT, was cloned into a pET28a vector with an N-terminal hexa-His-tag and an HRV14 3C protease cleavage site. Quick Change site-directed mutagenesis [23] was used to introduce three other mutations: C280S, H507D and S584K to obtain the CSH mutant. These plasmids were transformed into BL21 DE3 RIL *Escherichia coli (E.coli)* strain (Agilent Technologies, Santa Clara, USA). Expression of the proteins was carried out by inoculating 1 L of LB media containing 0.5% glycerol, 50 µg/mL of kanamycin, and 34 µg/mL of chloramphenicol with 50 mL of overnight LB cultures. Cells were allowed to grow at 37 °C until an OD of ~0.7 and transferred to 15 °C and grown until an OD of approximately 1. Media was supplemented with 20 mM MgSO_4_ or MgCl_2_ and expression induced with 1–2 mM IPTG and allowed to grow overnight at 15 °C overnight.

Cells were harvested and spun down at 5000× *g* for 25 min. Cell pellet was resuspended in 100 mL of lysis buffer (50 mM phosphate pH 7.6, 300 mM NaCl, 1 mM tris (2-carboxyethyl) phosphine (TCEP), 20 mM imidazole, 10% glycerol, 0.5 mM ethylenediaminetetraacetic acid (EDTA), 1 mM phenylmethylsulfonyl fluoride (PMSF)). Cells were sonicated for 10 min and spun down at 18,000× *g* for 30 min after which it was loaded onto a nickel gravity column pre-equilibrated with the lysis buffer. After loading, the column was washed with at least 10 column volumes of the lysis buffer followed by another 15 column volumes of a wash buffer (50 mM phosphate pH 7.6, 500 mM NaCl, 1 mM TCEP, 40 mM imidazole, 5% glycerol, 0.5 mM EDTA, 1 mM PMSF). The protein was eluted from the column using a phosphate buffer (50 mM phosphate pH 7.6, 300 mM NaCl, 300 mM imidazole, 5% glycerol, 0.5 mM EDTA). Eluted protein was either dialyzed overnight concurrently with the hexa-His-tag cleavage with HRV14 3C protease at a ratio of 1:20 cleavage reaction in excess of Tris-Cl loading buffer (20 mM Tris, 100 mM NaCl, 0.25 mM EDTA, pH 8.0) or diluted two-fold and incubated with HRV14 3C protease overnight at 4 °C.

The dialyzed protein was spun down and supplemented with 1 mM TCEP, loaded onto a heparin column pre-equilibrated with the Tris-Cl buffer as used for the dialysis but containing 1 mM TCEP. The column was subsequently washed with at least 15 column volumes of loading buffer followed by another 15 column volumes wash with the same buffer containing 300–500 mM NaCl. Elution of the protein from the heparin column was carried out in a single step using the Tris-Cl loading buffer containing 1.0 M NaCl and subsequently dialyzed against Tris-Cl storage buffer (20 mM Tris-Cl pH 7.4, 75 mM NaCl), concentrated to 25 mg/mL, snap frozen and stored at −80 °C. Purity of protein at each step was accessed by reducing SDS-PAGE gel.

### 2.2. Selenomethionine (SeMet) Labeling

A volume of 500 mL LB media was inoculated with a glycerol stock of CSH (BL21 DE3 RIL) and grown overnight in the presence of 100 mg of methionine, 50 µg/mL of kanamycin and 34 µg/mL of chloramphenicol. The cells were then pelleted at 4000× *g* for 10 min, washed with 20 mL minimal media and re-suspended in 1 L of minimal media supplemented with 1 mM MgCl_2_. The re-suspended cells were grown at 15 °C for 30–45 min in the presence of 50 mg each of isoleucine and leucine and 1000 mg each of SeMet, phenylalanine, lysine, and threonine before induction with 1 mM IPTG overnight. The cells were harvested and purified and stored the same way as unlabeled protein. The degree of SeMet incorporation was determined to be 89% using MALDI-TOF mass spectrometry (Appendix A).

### 2.3. Crystallization

Initial crystallization screening was carried out using commercially purchased precipitants in sitting-drop vapor diffusion and hanging-drop vapor diffusion experiments at a CSH protein construct at a concentration of 15–20 mg/mL at 4 and 20 °C with the help of a crystallization robot. Microcrystal hits were obtained within 48 h at both temperatures in Hampton Research (Aliso Viejo, CA, USA) Natrix HT E8 (0.05 M potassium chloride, 0.05 M sodium cacodylate trihydrate pH 6.0, 10% *w*/*v* polyethylene glycol 8000, 0.0005 M spermine, 0.0005 M L-argininamide dihydrochloride) and F9 (0.012 M sodium chloride, 0.08 M potassium chloride, 0.04 M sodium cacodylate trihydrate pH 6.0, 50% *v*/*v* (+/−)-2-methyl-2,4-pentanediol, 0.012 M spermine tetrahydrochloride). Optimization was carried out by changing pH of precipitant, protein concentration, and variation of the concentrations of the individual components in a precipitant as well as the protein: precipitant ratio. None of these improved the crystal form significantly for structure determination (Appendix A). The Hampton additive screen was carried out using the Natrix conditions E8 and F8. Crystals grown in 30 mM glycylglycylglycine (GGG)-C11, and 10 mM EDTA-E1 at 20 °C were of slightly different morphology than the initial crystal hit. Further optimization was therefore carried out using glycylglycine and EDTA as additives by varying their concentrations. These conditions produced thin plates of less than 5 µm thickness in each of the individual cases in the presence of 200 mM glycylglycine or 100 mM EDTA which diffracted X-rays to approximately 4.5–5 Å resolution. The choice of glycylglycine (GG) was serendipitous as it was available in the lab at the time, whereas GGG was not. Crystals grown with GGG when it was finally obtained diffracted more poorly.

Even though the resolution of these datasets was sufficient in theory to solve the structure, the diffraction was highly anisotropic with very high mosaicity, making it practically impossible to solve the structure using these datasets. A 1:1 mixture of the two conditions containing GG and EDTA was carried out since the path to further optimization had been narrowed much further. Surprisingly, the crystals grown in these conditions were significantly thicker than those produced by their individual conditions. Thus, GG and EDTA together had a cumulatively positive effect on the crystal size even though they remained plates emanating from a single nucleation site (Appendix A).

The following conditions produced the crystals that were suitable for diffraction studies: 50 mM KCl, 50 mM sodium cacodylate trihydrate pH 6.0, 12% PEG 8000, 1.0 mM spermine, 1.0 mM L-argininamide, 200 mM glycyglycine or glycylglycylglycine, and 50 mM EDTA in a sitting-drop vapor diffusion crystallization format between 10 and 25 °C with drop sizes ranging between 1 and 10 µL in micro bridges (Hampton Research, Aliso Viejo, CA, USA). The EDTA could be substituted with 10 mM MgCl_2_, 10 mM MnCl_2_, or 100 mM CaCl_2_ with similar results. The wild-type crystals as well as SeMet CSH crystals were grown with 100 mM CaCl_2_ instead of EDTA and 2–5 mM TCEP supplemented with the protein. The CSH mutant of PFV routinely formed plate-like crystals, which diffracted X-rays to at least 3.0 Å resolution, enabling structural solution. Optimized crystallization conditions using this mutant were subsequently used to crystallize and solve the structure of the WT as well.

Crystals were harvested using MiTeGen (Ithaca, NY, USA) MicroLoops or MicroMesh mounts and cryo-protected in reservoir supplemented with 25% glycerol before flash freezing.

### 2.4. Data Collection and Processing

X-ray diffraction data were collected at the Cornell High Energy Synchrotron Source (CHESS) F1 beamline, the Advanced Photon Source (APS) 23-ID-D beamline, and the Stanford Synchrotron Radiation Light Source (SSRL) 9-2 beamline. Both PFV PR-RT WT and CSH crystallized with C2 space group symmetry. Data processing and scaling were performed using iMosFlm and Aimless. The structure was solved by merging five SeMet-labeled data sets of the CSH mutant using Crank2, and the refinement was carried using REFMAC, all in the CCP4i suite [24]. Model building was performed in Coot [25]. The model of the SeMet-labelled CSH mutant was successfully used for molecular replacement using the WT data which was found to be twinned (using program Xtriage) in Phaser in PHENIX [26] and refined to 2.9 Å resolution using the twin law -h, -k, l to R-work/R-free of 22.4/25.8% in REFMAC [27] and phenix.refine [26,27]. Publication figures were generated using PyMOL Molecular Graphics System, v. 1.7 (Schrödinger, LLC, New York, NY, USA) and https://biorender.com/ (accessed on 6 January 2021).

### 2.5. Modeling of the PFV PR Dimer Structure

Two PR-RT structures were positioned across a two-fold symmetry axis to generate a dimeric PR. The model geometry was regularized using the phenix.geometry_minimization tool until convergence. MolProbity (http://molprobity.biochem.duke.edu/ (accessed on 5 May 2021) was used for model validation. The MolProbity summarized statistics are included in the Appendix A, as well as the coordinates for the in silico PFV PR dimer. cealign in PyMOL Molecular Graphics System, v. 1.7 (Schrödinger, LLC) was used to align to the structure of apo HIV-1 PR (PDB ID 2HB4), with an overall RMSD of 3.11 Å relative to the PFV PR dimer model.

### 2.6. Extension and Processivity Assay

The polymerase and RNase H activity assays were carried out according to the protocol reported in [28,29].

### 2.7. Protein–DNA Cross-Linking

The cross-linking experiments were conducted as reported in [30,31,32]. NaCl concentrations in the final reactions were, however, changed to 150–200 mM.

### 2.8. Accession Number(s)

The coordinates and structure factors for the crystal structures of Prototype Foamy Virus Protease-Reverse Transcriptase CSH mutant (SeMet-labeled) and native were deposited in the Protein Data Bank (PDB) under accession numbers 7KSE and 7KSF, respectively.

## 3. Results

### 3.1. Engineering a Proteolytically Resistant Mutant of PFV PR-RT for Bacterial Expression

Wild-type PFV PR-RT is highly susceptible to proteolysis when expressed in bacterial strains, as observed in Boyer et al. 2004 [28]. The breakdown products complicated the development of purification schemes that result in pure protein, which is required for structural and biophysical studies. Instability of proteins due to their primary sequence is a longstanding problem in biology [33]; this problem is commonly seen for flexible polyproteins. In an effort to improve the proteolytic stability of PFV PR-RT, which has a protease null mutation (D24A), and is designated wild type (WT), in silico mutagenesis of WT PFV PR-RT was carried out by replacing each amino acid with the other 19 and computing an instability index on the ExPASy ProtParam server (http://web.expasy.org/protparam/ (accessed on 25 October 2014) using a Python script (see Appendix A).

The instability index computed is a weighted sum of the dipeptide composition of proteins that is based on a strong correlation between protein stability and the dipeptide composition [33]. Proteins with instability index estimates below 40 are considered stable while those above that threshold are considered unstable [33]. The instability index for the WT protein was computed to be 39.75, very close to the upper limit of the stability threshold. H507 and S584 were identified as potential instability hotspots in the primary sequence of PFV PR-RT. The calculation suggested that substituting several other amino acids at these positions would help stabilize the protein. Two mutations, H507D and S584K, were made in the PR-RT. As discussed below, these two residues have no significant impact on the structure and activity of the protein. In addition, we also made the C280S mutation to improve the solution behavior of the protein akin to HIV-1 RT C280S, which is known to improve the behavior of purified HIV-1 RT (it is a coincidence that these two residues, which are located in different subdomains in the respective proteins, have the same amino acid number, 280; they are not positional equivalents). The PR-RT mutant C280S/H507D/S584K, designated CSH, gave a 4-fold increase in expression compared to the WT in bacterial strains. This construct could be purified in two steps (nickel affinity and heparin) to at least 95% purity in the absence of protease inhibitors and reducing agents (Appendix A).

Changes in the sequence of a protein can have unintended consequences on the structure and/or the activity of the protein. To determine whether the mutations we made in PR-RT have an effect on the enzymatic activities of the protein, the polymerase, processivity and the RNase H activities were assayed for both the WT protein and the CSH mutant. The WT enzyme showed robust polymerase, processivity, and ribonuclease activity (Appendix A) that was consistent with data published in Boyer et al. 2004 [28]. However, since the PR-RT has a mutation that inactivates PR, D24A, proteolytic activity was not determined. When compared to WT, the CSH mutant had similar RNase H activity and showed a modest reduction in the polymerase activity (Appendix A). The processivity of the polymerase was reduced approximately 3-fold (Appendix A) compared to the WT. It is possible that replacing H507 with an aspartic acid, which substitutes a negative charge for a positive charge, could affect nucleic acid binding.

### 3.2. Architecture of PFV PR-RT

The higher purity and greater yield of the CSH mutant of PFV PR-RT made it possible to use broad robotic crystallization screens, which resulted in the growth of crystals that diffracted to ~3 Å resolution. The same crystallization condition that yielded the crystals of the CSH mutant did not produce usable crystals of the WT protein; relatively fragile, but usable, crystals of WT protein could be produced by modifying the crystallization conditions (see Methods). Both the CSH mutant and WT PFV PR-RT crystallize as monomers with one copy in the asymmetric unit in space group C2. Despite differences in the unit-cell parameters, particularly along the x-axis, the structures of CSH and WT PR-RTs are highly superimposable (RMSD = 0.785 Å). Unless specifically indicated, the results we discuss here are from the WT protein structure. The structure of the WT PR-RT was initially solved by using the anomalous signal from the selenomethionine (SeMet)-substituted CSH mutant, and the model obtained was used as the search model for molecular replacement using the WT data (Table 1 and Figure 1). Even though the sequence homology of PFV and other retroviruses is generally low, typically less than 30%, the individual structural elements of retroviral polymerases and proteases are highly conserved (Figure 1). There are notable differences in terms of the relative positions of the domains which are described in the subsequent sections.

### 3.3. PR and PR-CTE

As a prototype aspartyl protease, HIV-1 PR contains four main structural regions: the dimer interface (residues 1–10 and 90–99), the core domain, which is present in each monomer (residues 11–22, 29–39, 61–80, and 85–90), the active site (residues 23–28 and 81–84) containing the conserved catalytic DTG triad, and the flap region (residues 40–60) [34]. The PFV PR has a closed barrel-like core domain constituted predominantly of β sheets that is quite similar to one half of the dimeric PR of HIV-1 [12]. The monomeric PFV PR is composed of seven β strands (β1^PR^–β7^PR^), two short helices (αA^PR^ and αB^PR^), and random coils (Figure 1 and Figure 2A). Although the PFV PR monomer is well folded in the PR-RT polyprotein, it must form a homodimer to function as a protease [18].

Residues 1–4 in the PFV PR monomer, which form a part of the putative dimer interface in active PR (inferred from structures of mature HIV-1 PR), had no defined electron density and residues 5–9 had weak electron density (Figure 2A). Strands β1^PR^ and β2^PR^ (residues 12–15 and 20–25), which form the “fulcrum”, are connected by a hairpin loop A1 (residues, 16–19), the so-called “10′s loop” [35]. The continuous hairpin loop, which contains the conserved DSG catalytic triad (residues 24–26) at the active site, is bordered on each side by β2^PR^ and β3^PR^(residues 35–35) in the “fireman’s grip.” This large loop is tilted slightly relative to the core and points away from the RT, which exposes the active site to solvent. An α-helix (αA^PR^, residues 36–38) present in a typical pepsin-like PR fold, is extended by a loop that forms the “flap elbow” of the PR following β3^PR^ (Figure 2B,C). The presence of this helix is one of the ways PFV PR differs from HIV-1 PR, which has a large loop instead of a helix in this region (Figure 2C). This loop extends to β4^PR^ (residues 45–49) and β5^PR^ (residues 58–68) which constitute the “flap.”

The hairpin loop at the junction of β4^PR^ and β5^PR^ forms the tip of the flap (residues 50–57) which opens and closes to allow the binding and release of the substrate. Consistent with its function, this region is partially disordered and has weaker electron density in the monomeric PFV PR structure. β5^PR^ (residues 58–68), which traverses almost the entire PR core, is connected by a turn (residues 69–70) at its C-terminus to β6 ^R^ (residues 71–80). This turn, which together with the 10′s loop, defines the PR “exosite” and serves as a hinge which, together with the mid-section of β5^PR^, allows the tip of the flap to open and close (24). The β6^PR^ strand is twisted away from β5^PR^ in a way that creates a gap between the flap region and the active site loop. When PFV PR dimerizes, this gap will form the substrate-binding cavity (Figure 2C). The large P1-loop, which lines the walls of the active site, connects β7^PR^ (residues 85–87) and β6^PR^. β6^PR^ and β3^PR^ form the only parallel β sheet in the structure.

A short coil and a short helix (half-helical turn, αB’) link the C-terminus of β7^PR^ to a long random coil that extends beyond the C-terminus of the PR domain. The only helix in HIV-1 PR is at the C-terminus. The C-terminus of PFV PR, which together with the N-terminus, forms a major portion of the dimerization interface, is also a random coil in our crystal structure. It is not known whether, upon dimerization of PFV PR, these would form β sheets, as they do in the HIV-1 PR dimer. The first 100 residues of PFV PR-RT are sufficient to define the structure of the entire PR monomer as is the case for HIV-1 PR [36].

Additional residues at the C-terminus of PFV PR have been identified in multiple sequence alignments (Appendix A). Residues 101–145, which were not recognized as being part of either PR or RT, have been designated as the protease C-terminal extension (PR-CTE) (Figure 1 and Figure 3). Although HIV-1 PR does not have a similar C-terminal extension, other retroviruses, including Moloney murine leukemia virus (MoMLV), mouse mammary tumor virus (MMTV), and Mason-Pfizer monkey virus (MPMV) have CTEs of varying lengths (Appendix A). The PFV CTE contains a long linker that stretches from the C-terminus of the PR to the N-terminus of RT, which is composed of three helices (designated αC^CTE^, αD^CTE^, and αE^CTE^) that pack against helices F, G, and strand 4 of RT (Figure 1 and Figure 3). These three structural elements of PFV RT are the equivalents of helices E and F and β strand 6 in HIV-1 RT, which form part of the “floor” of the palm subdomain that contains the polymerase active site. The combined PR CTE helices and PFV RT helices F, G, and strand 4 create an extensive hydrophobic network that extends the floor of the palm. This overall network allows the PR monomer to bind to the RT domain close to the fingers subdomain. This arrangement would allow PR to swing or tilt in ways that could enable dimerization without any anticipated steric interference; the linker is long enough to allow the PR to be displaced from RT although PR and RT would still be covalently connected.

We positioned two PR-RT structures in silico to generate a dimeric PR in PyMOL followed by energy minimization using the phenix.geometry_minimization tool [26] until convergence was reached. We were able to produce a PR dimer that was poised for catalysis without significant steric clashes with the RT (Figure 2B,C), suggesting that this structure could be proteolytically active without any major structural rearrangements of either PR or RT. While this model does not offer specific insights into how dimeric interactions of PR might involve or affect RT, the model offers a glimpse of a plausible mechanism for PR dimerization and suggests how dimeric PR could be formed by two Pol polyproteins. The substrate-binding groove of PFV PR generated in silico appears to be wider than in HIV-1 PR (Figure 2B,C) but is likely to adopt a similar type of groove when binding the peptide segments that would allow PR-RT to process the Gag C-terminal 3 kDa peptide and RT/IN junctions during virion maturation. It is worth noting that HIV-1 PR inhibitors such as tipranavir, darunavir, and indinavir [18] do not inhibit PFV PR.

### 3.4. RT Domains and Subdomains

Due to their resemblance to a right hand, the subdomains in the structures of polymerases, including retroviral RTs, have been named after the segments of a right hand.

The arrangement of the subdomains of the Pol portion of our PR-RT structure does not resemble a right hand but instead is similar to the configuration of HIV-1 RT p51. The hand-like active conformation of the Pol portion of HIV-1 RT p66 is stabilized by the compact rearrangement of subdomains (fingers, palm, thumb, and connection) in p51. In contrast, the monomeric PFV RT is in a compact conformation that would have to undergo an extensive rearrangement of the subdomains into the ‘right hand’ configuration to function as a polymerase. However, all of the subdomains observed in other retroviral RTs—fingers, palm, thumb, connection, and the RNase H domain—are easily recognizable (Figure 1 and Figure 3A) even though their relative positions are different. This rearrangement of PFV RT from a catalytically inactive to an active configuration may be facilitated by relatively ‘weak’ interactions between subdomains. It is possible that the energy barrier for isomerization from a p51-like to a p66-like configuration is lower for PFV RT, compared to HIV-1 p51, where the interfaces are predominantly hydrophobic [37]. The details of these interactions are discussed below.

### 3.5. Fingers and Palm Subdomains

The fingers subdomain of PFV PR-RT (144–228, 260–290) is characterized by four helices (αA, αB, αD, and αE), five β strands (β1–β5), and random coils. The N-terminus of PFV RT is part of a long loop that ends at the beginning of helix A, which is similar to HIV-1 and MoMLV RTs. Three anti-parallel β sheets are sandwiched by three helices to form the hydrophobic core of the fingers, which is similar to other RTs. β2 and β3 define a hairpin loop that points towards the putative nucleic acid binding cleft, which is similar to the β3–β4 loop in HIV-1 RT p66; the β3–β4 loop (see https://hivdb.stanford.edu/pages/3DStructures/rt.html#RT_p66 (accessed on 29 June 2021) for HIV-1 RT secondary structure annotation) has been implicated in dNTP binding [10,38] (Figure 1 and Figure 3). The palm of the PFV PR-RT (229–259, 291–367), which is surrounded by the connection and fingers subdomains, and the PR-CTE, comprises three helices and four anti-parallel β sheets, and is similar to other RTs (Figure 1 and Figure 3). Three highly conserved β sheets, which harbor the conserved catalytic motif YVDD (314–317), constitute the polymerase active site (Figure 1 and Figure 4A). In PFV RT, the primer grip (355–368), which is a short β-hairpin in other RTs [10], is a somewhat large loop that points towards the active site hairpin loop. The amino acid residues that form the primer grip hairpin loop are generally conserved in PFV (LGF), MoMLV (LGY), XMRV (LGY), and HIV-1 RT (229–231, MGY) (Appendix A). We suggest that the flexible primer grip of PFV RT could reduce the constraints imposed by more rigid β sheets in other RTs, which could help orient the primer during nucleic acid synthesis and could contribute to the higher processivity of PFV compared to other RTs.

The architecture of the active site of PFV RT is similar to other RTs. The polymerase active site loop formed by β7 and β8, carries the catalytic aspartates D316 and D317; these residues correspond to D185 and D186 in HIV-1 RT p66. The catalytic aspartate triad is completed by the conserved residue, D254, situated at the tip of β4 which sits upright next to β7 and β8. The corresponding residue in HIV-1 RT p66 is D110 (Figure 4A) [38,39]. The PFV RT mutant V315M, which is in a position that corresponds to amino acid M184 in the active site of HIV-1 RT p66, leads to 50% loss of polymerase activity in virions, with no observable full-length cDNA detectable in transfected cells and a greatly reduced processivity of the purified enzyme [28,29]. In contrast, in HIV-1 M184V is an efficient polymerase; however, the M184V mutant allows HIV-1 RT to discriminate against L-nucleoside analogs.

Helix F of PFV RT (helix E in HIV-1 RT p66) is oriented ~100° relative to helix G (helix F in HIV-1 RT p66) above the active site. Helix G of PFV RT (helix F in HIV-1 RT p66) packs against strands β4 and β8 in a way that is similar to that of HIV-1 RT p66 (Figure 1). The fingers and palm subdomains are in the same relative positions in PFV RT as in other RTs.

### 3.6. Thumb and Connection Subdomains

The thumb subdomain of PFV RT (378–449), which is characterized by a three-helix bundle, has moved away from the palm and is positioned next to the connection subdomain (450–590), making extensive contacts with it (Figure 1, Figure 3, and Figure 4D). The connection subdomain, characterized by a five-stranded mixed β sheet that is stabilized by five helices, lies between the fingers/palm on one side and the RNase H/thumb on the opposite side. The connection subdomain is further tilted slightly towards the polymerase active site such that it sits almost parallel to the palm of the RT compared to its position in the p66 subunit of HIV-1 RT where the two subdomains are orthogonal (Figure 3C). This compact fold allows helix L (K in HIV-1 RT) to pack against H, while helix O (L in HIV-1 RT) tilts towards the active site loop positioning T552 of PFV RT to form a hydrogen bond with Y314, and T556 with H236, while H507 forms a bifurcated hydrogen bond with S237 and N364 (Figure 4B). The H507D mutant employed in the SeMet protein also makes analogous contacts, and the SeMet PR-RT crystallizes much faster than WT. The H507F/I mutants, which lose these hydrogen-bonding interactions, did not crystallize despite several attempts. This underscores the importance of these specific hydrogen bonds in stabilizing the conformation of PFV PR-RT. The C-terminus of helix L also makes contacts with the primer grip and helix C of the palm (Figure 4A).

The folding of the connection subdomain onto the palm leads to significant interactions between the palm and thumb in a way that partially blocks access to the polymerase active site. There is little interaction, however, between the connection and the fingers subdomains. This is in stark contrast to the structure of the p51 subunit of HIV-1 RT, where the positioning of helix L and β-strand 20 allows the loop that connects them to make close contact with the fingers (Figure 3B and [28]). Weak interactions between the connection and fingers in PFV RT would permit a rearrangement of the subdomains which would allow PFV RT to attain the polymerase-competent conformation; the connection and fingers do not interact in HIV-1 p66 (Figure 3C).

The three-helix bundle that is characteristic of the thumb (αH, αI, αJ), together with helix L, sandwiches three β sheets (9,12,13) of the connection domain, concomitantly forming an extensive interface through a mixture of hydrophobic and van der Waals interactions and hydrogen bonds (Figure 1, Figure 3A and Figure 5C). In the p66 subunit of HIV-1 RT, antiparallel β sheets at the base of the thumb help to position the thumb. The corresponding amino acids of PFV RT are random coils, and the thumb is in an extended position, and is parallel and adjacent to the connection subdomain. This configuration of the thumb of PFV RT is stabilized by its interactions with the connection subdomain and the RNase H domain. In the p51 subunit of HIV-1 RT (Figure 3B), the thumb is extended away from the connection subdomain and has only minimal interactions with it, which allows helix H to interact extensively with the RNase H domain of the p66 subunit. Compared to the position of the thumb of PFV PR-RT, this represents close to a 90° rotation (Figure 3A,B). Some of the interactions at the connection/palm and connection/thumb interfaces are shown in Appendix A.

### 3.7. RNase H

The RNase H domain of PFV RT (593–751) consists of an asymmetric arrangement of five-stranded mixed β sheets and four α-helices with the β sheets sandwiched between a three-helix bundle formed by helices αA^RH^, αB^RH^, αC^RH^ and αD^RH^ (Figure 1 and Figure 5B). There is a long C-terminal helix, αE^RH^_,_ which traverses the entire length of the RNase H domain on a side facing the solvent. The RNase H domain is swiveled around the connection and the thumb subdomains, and is positioned next to the fingers, opposite the connection/thumb subdomains (Figure 3 and Figure 4D). While the overall architecture is similar to the structures of other RNases H, the PFV PR-RT RNase H domain generally has longer β-strands and helices compared to the RNase H domain of HIV-1 RT (Figure 5A). N687 serves a hinge between helices αB^RH^ and αC^RH^ forming a continuous kinked helix, which is oriented across helices αA^RH^ and αD^RH^. Although it is absent in HIV-1, αC^RH^ (also referred to as the C-helix) is present in *E. coli* RNase H, human RNase H, and MoMLV RNase H, and is, in these RNases H, critical for binding substrates and for enzymatic activity [40]. αB^RH^ and αC^RH^ are positioned such that, while αB^RH^ packs across helices αA^RH^ and αD^RH^ and helps to stabilize them, αC^RH^ is positioned across the thumb and the connection, with which it forms van der Waals contacts that help to restrict the motion of the thumb (Figure 1 and Figure 5B,C). In the WT PFV PR-RT structure, there is a calcium ion from the crystallization buffer (which contains 100 mM Ca^2+^) that is chelated by the side chains of residues D601, E648, and D671, which define the RNase H active site (Figure 5B). These residues are positional equivalents of D443, E478, and D498 of HIV-1 RNase H. In the PFV PR-RT structure, the RNase H domain occupies the opposite side of the DNA-binding cleft (Figure 4D), with the active site pointing away from the putative nucleic acid-binding cleft. This domain must undergo a rotation of 180° to bring the active site into register with a bound RNA/DNA substrate.

### 3.8. Buried Surface Analysis of Interactions among the PR-RT Subdomains

The conformation of RT in the structure of PFV PR-RT cannot be active, and the structure has to undergo significant conformational rearrangements to be able to bind to a nucleic acid substrate. Buried surface analysis of macromolecular complexes and interfaces among the (sub)domains can provide a glimpse of the energetic cost required for the conformational rearrangements of various structural/functional states of a multidomain proteins and/or complexes. The amount of buried surface area has a direct correlation with the binding affinity and the dissociation constant (K_D_) of the complexes but corresponding ΔG values are dependent on the nature of interactions involved at the interfaces [37,41,42].

Buried surface calculations for the PFV PR-RT were carried out using the ccp4 online program jsPISA [43,44]. The connection-thumb interaction buries 1216 Å^2^ (with corresponding ΔG of interaction estimated at −17.9 kcal/mol) (Table 2), which is the largest intersubdomain interface, while the RNase H-thumb interface buries only 164 Å^2^ (−2.0 kcal/mol). Interdomain interactions among PR, polymerase, and RNase H are less extensive than the intersubdomain interactions within RT. This suggests that the positioning of PR, polymerase, and RNase H domains relative to each could be highly adaptive. The interacting surface area of the RNase H with the thumb, which would dissociate when PFV PR-RT assumes an active conformation, is only 164 Å^2^ (−2.0 kcal/mol); this interaction is between the C^RH^-helix and the thumb. A total of 425 Å^2^ (−3.3 kcal/mol) is buried between the connection and the RNase H. The sizes of these interfaces suggest that the interaction between the RNase H and the rest of the RT is relatively weak and could permit the rotation of RNase H relative to the rest of RT to allow binding to nucleic acid. The PFV connection subdomain, which is nestled between the fingers, palm, and thumb subdomains has extensive contacts with other subdomains of RT. The sum of all of the buried surface area involving the connection subdomain is greater than 2000 Å^2^ (~−30 kcal/mol) (Table 2). Because this subdomain forms part of the nucleic acid binding cleft, significant rearrangements are required to permit nucleic acid binding. There is a progressive increase in the buried surface area and their corresponding ΔG values between the connection and the fingers, RNase H, palm, and thumb, respectively. This would allow for a sampling of different conformational states that could facilitate nucleic acid binding. The PR interface with the fingers subdomain buries ~630 Å^2^ of the surface area (−7.5 kcal/mol). While this interface may stabilize interactions between PR and RT, there should be sufficient flexibility to permit the dimeric PR to flex to access different substrates during processing. On the other hand, the extensive interaction between the PR CTE and the palm subdomain (~1000 Å^2^, −17.4 kcal/mol), helps to constrain the relative positions of the PR and RT domains.

### 3.9. PFV PR-RT Complexes with Nucleic Acids

We measured the dissociation of PFV PR-RT from an RNA/DNA template-primer and a double-stranded (ds) DNA and compared the binding of these substrates to the binding of HIV-1 RT. PFV PR-RT dissociated much more slowly when bound to dsDNA than from an RNA/DNA hybrid. In contrast, HIV-1 RT dissociated much more slowly from an RNA/DNA hybrid than from a dsDNA (Appendix A). This suggests that PFV and HIV RTs interact differently with their nucleic acid substrates, and the differences in the interactions may be relevant for understanding the differences in processivity between these two RTs.

To probe the similarities between the nucleic acid binding clefts of PFV PR-RT and that of HIV-1 RT, a site-specific DNA cross-linking experiment was carried out. The Q258C substitution in HIV-1 RT makes it possible to covalently trap appropriately modified nucleic acid substrates in the binding groove of the RT. The chemistry that was designed to covalently cross-link HIV-1 RT to dsDNA or to RNA/DNA is efficient with a single round of nucleotide incorporation followed by translocation. This allows a thioalkyl-modified guanine base G (at the N2 position) in the primer to be positioned in register with the mutated residue C258, which permits the cross-linking disulfide bond to form [32,45]. The covalently trapped complex can be purified using tandem nickel-heparin columns. PFV PR-RT Q391C, which is the equivalent of HIV-1 RT Q258C, based on the sequence alignments (Appendix A), can be used to covalently trap appropriately modified dsDNA in a reaction that is similar to what has been performed with the Q258C HIV-1 RT mutant (Appendix A). Thus, even though PFV RT Q391 in the thumb is pointing away from the putative nucleic acid binding cleft and exposed to solvent in the structure we report here, upon rearrangement, the thumb of PFV RT is positioned in a configuration similar to the thumb of HIV-1 RT p66, and at a distance from the polymerase active site that is equivalent to that of the thumb in p66. The successful cross-linking suggests that the overall configuration of the catalytically active PFV RT is similar to that of the p66 subunit of HIV-1 RT.

## 4. Discussion

Retroviral genomes encode aspartyl PRs that are responsible for the proteolytic processing of polyprotein precursors and RTs that carry out the conversion of the single-stranded RNA genome into a dsDNA, which is subsequently inserted into the host chromosome by IN. In the crystal structure of PFV PR-RT reported here, both PR and RT are monomeric. PR is folded similarly to a single subunit of the mature dimeric HIV-1 PR. The N- and C-terminal residues that would be involved in the dimerization interface of the active form of PR are random coils in the PFV PR monomer. In silico placement of two PFV PR-RT molecules (related by two-fold symmetry) can be used to model the structure of an active PFV PR dimer, without significant steric clashes in either the PR or the RT domains. This offers a glimpse of how this PFV PR-RT might be able to carry out proteolytic processing of peptide substrates.

On the other hand, the polymerase domain in the current structure is in a configuration that cannot bind nucleic acid, and more closely resembles the p51 subunit of HIV-1 RT, which plays a structural role in the mature p66/p51 heterodimer. This implies that a significant rearrangement of the subdomains in RT must occur before PFV PR-RT can bind a nucleic acid substrate. There are long linkers that connect the palm and the thumb, and the connection and RNase H, which would allow the proposed movements to occur (Movie S1). Although the proposed rearrangement of the subdomains of the monomeric form of PFV RTs may appear to be unusual, the rearrangements may not be unique to the PFV RT. Based on an analysis of buried surface areas in the HIV-1 p66 and p51 heterodimer, it has been suggested that monomeric forms of these enzymes would adopt a more compact “p51-like” conformation [20]. Zheng et al. [22] used NMR spectroscopy to confirm this initial observation, further asserting that the RNase H domain of monomeric HIV-1 p66 has a loose structure with flexible linkers that connect the thumb and RNase H domains to the rest of the structure, without any significant interaction between the thumb and the RNase H.

This is consistent with buried surface area calculations of PFV PR-RT which show that only approximately 164 Å^2^ of the surface is buried between the RNase H domain and the thumb corresponding to −2.0 kcal/mol, while 425 Å^2^ is buried between the RNase H and the connection subdomain which translates to −3.3 kcal/mol. This interaction between RNase H and the connection/thumb subdomains involves the C-helix of PFV RNase H. HIV-1 RT lacks the C-helix; thus, the interactions observed between the thumb and RNase H of PFV RT do not occur in HIV-1 RT. Although it is not clear whether, for HIV-1 RT, a monomeric p66 subunit is relevant to the maturation of HIV-1 polyproteins in vivo, it has been shown in vitro that monomeric HIV-1 p66 predominantly samples two conformational states prior to dimerization and subsequent cleavage of the RNase H domain to produce the p51 subunit [21,22]. Transient sampling of a catalytically competent open conformer of p66 might expose the dimerization interface, leading to dimerization with a subunit in a “p51-like” conformer prior to maturation. It is possible that in HIV-1 an asymmetric homodimer precedes a subsequently matured asymmetric heterodimer. Structural data defining the position of the RNase H domain in the asymmetric homodimer prior to maturation has been elusive; neither the structure of a monomeric HIV-1 p66 nor a larger precursor that contains the second RNase H domain is available. The PFV PR-RT structure reveals what could be an isomerization intermediate of other RTs.

Superposition of the current structure on the HIV-1–RT–DNA complex (Figure 4C,D) suggests that the connection subdomain acts as the “gatekeeper”, by occupying the nucleic acid-binding cleft. For nucleic acid binding to occur, the connection subdomain must undergo close to a 90° rotation followed by a translation perpendicular to the helical axis of helix L (which is part of the connection subdomain). This screw movement would disrupt the interaction with the thumb, allowing it to move closer to the fingers/palm, and would push the RNase H domain out so that it would swivel to a position similar to what has been observed for other RTs. This would lead to a conformation for RT that would be relevant for nucleic acid binding (Movie S1). It is likely that the catalytically competent conformation is sampled in solution and is trapped and stabilized by binding to a nucleic acid substrate.

## 5. Conclusions

The structure of the PFV PR-RT polyprotein suggests molecular mechanisms by which PR and RT activities can occur, requiring PR dimerization for processing activity, and significant rearrangement for RT activity (summarized in Figure 6). In the case of the poliovirus 3CD polyprotein precursor, which contains the 3C protease and 3D RdRp in tandem, there is interplay between the two domains in which the RdRp activity is inhibited when 3C is present [46,47]. Conversely, in the case of PFV PR-RT, both activities must occur in the mature polyprotein form. As more information emerges regarding the structure and function of viral polyprotein precursors, we may learn additional themes about spatial and temporal regulation of viral assembly, maturation, and enzymatic function.

## Figures and Tables

**Figure 1 viruses-13-01495-f001:**
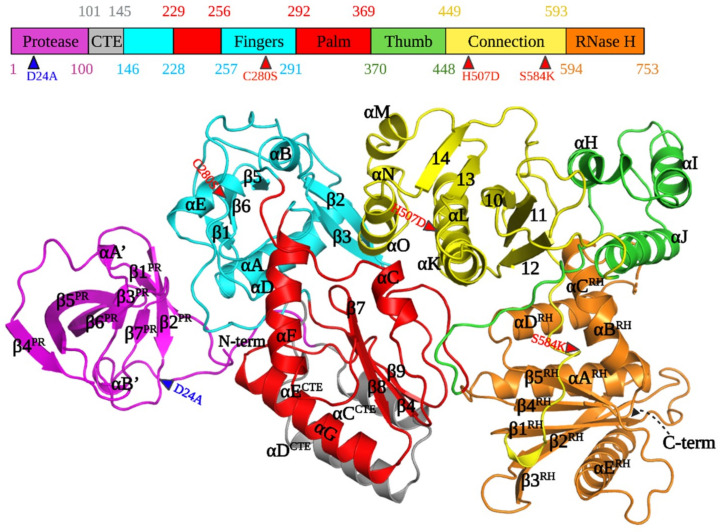
Annotated diagrams of the sequence and structure of PFV PR-RT. Alpha helices in the RT portion are labeled A-O and beta strands are numbered 1–14; CTE: C-terminal extension and RH: RNase H. A topological map of the secondary structural elements has been provided in Appendix A. HIV-1 RT secondary structure labeling can be found at: https://hivdb.stanford.edu/pages/3DStructures/rt.html#RT_p66 (accessed on 28 June 2021).

**Figure 2 viruses-13-01495-f002:**
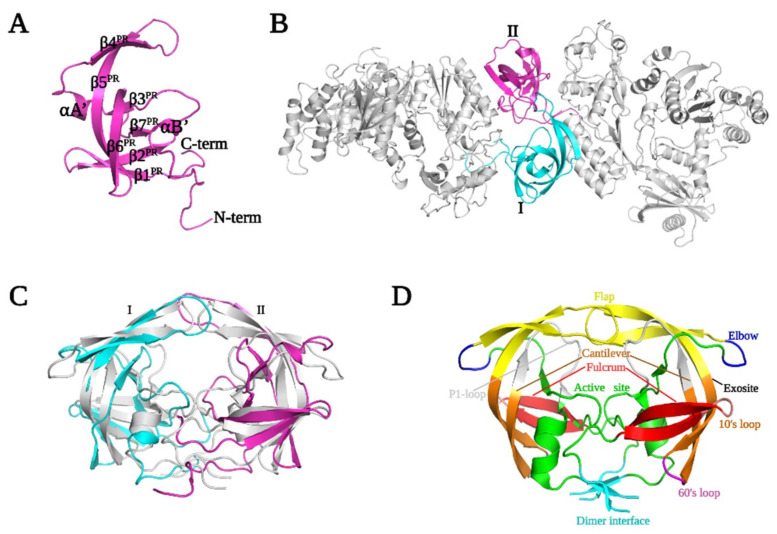
Model of a dimeric PFV PR. (**A**) Dimers of PR-RT generated in silico by a 2-fold rotation. (**B**) The monomeric PFV PR domain in our crystal structure. (**C**) Superposition of dimeric HIV-1 PR (PDB ID 2HB4 (grey) on dimeric PFV PR generated in silico by 2-fold rotation. (**D**) Annotated diagram of HIV-1 PR.

**Figure 3 viruses-13-01495-f003:**
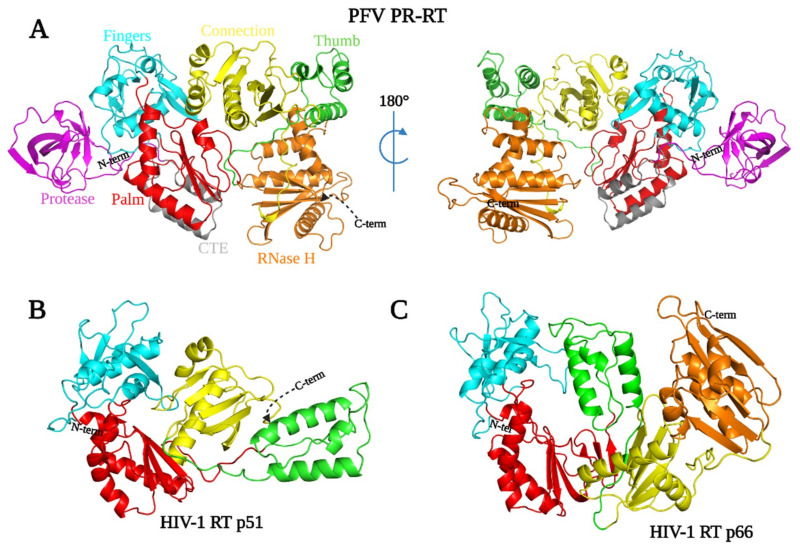
Structural arrangements of domains and subdomains of PFV RT and HIV-1 RT. (**A**) Architecture of PFV PR-RT colored by subdomain, oriented from the N-terminus to the C-terminus (top left panel); from C to N-terminus (top right panel). (**B**) HIV-1 RT p66, color-coded by subdomain. (**C**) HIV-1 p51. Equivalent subdomains in each drawing are color-coded as in panel A.

**Figure 4 viruses-13-01495-f004:**
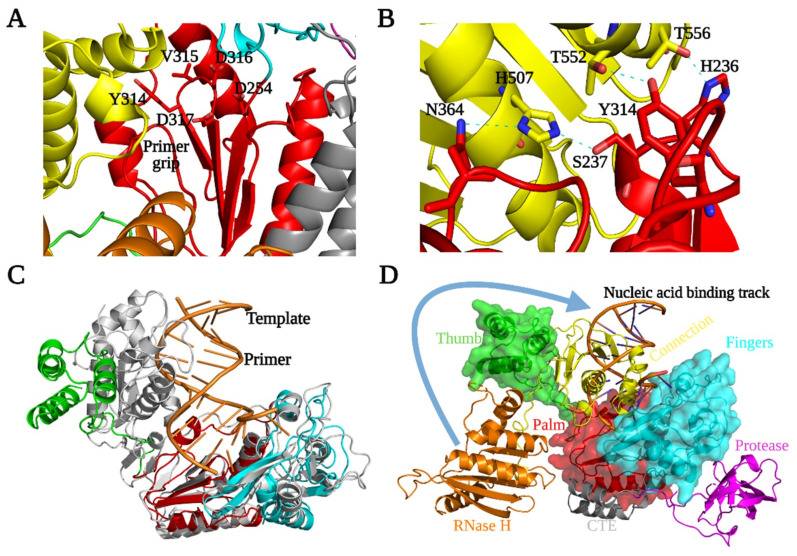
Structural arrangement of RT subdomains. (**A**) Key hydrogen-bonding interactions between connection and palm in PR-RT. (**B**) The PFV RT polymerase active site residues and some conserved structural elements found in retroviral RTs. (**C**) Structure of PFV PR-RT (subdomain coloring as in Figure 3) superimposed onto the HIV–1 RT–DNA complex (gray) (PDB ID 1N6Q). (**D**) Surface rendering of the fingers-palm-thumb subdomains in PFV PR-RT with the connection and RNase H backbones shown as ribbons.

**Figure 5 viruses-13-01495-f005:**
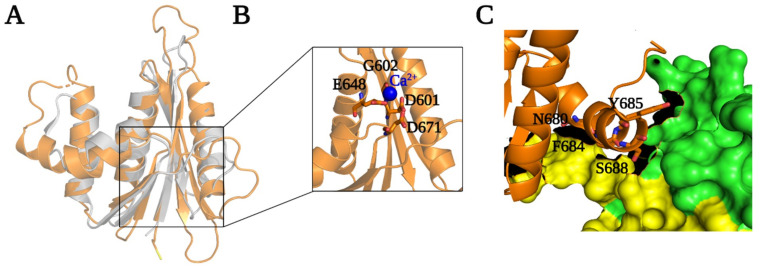
RNase H domain of PFV PR-RT. (**A**) Superposition of RNase H subdomains of PFV (orange) and HIV-1 (grey). (**B**) Active site of PFV PR-RT RNase H subdomain residues chelating Ca^2+^ (blue sphere). (**C**) Surface rendering of the interface between PR-RT thumb/connection subdomains and potential interactions with RNase H subdomain helix C.

**Figure 6 viruses-13-01495-f006:**
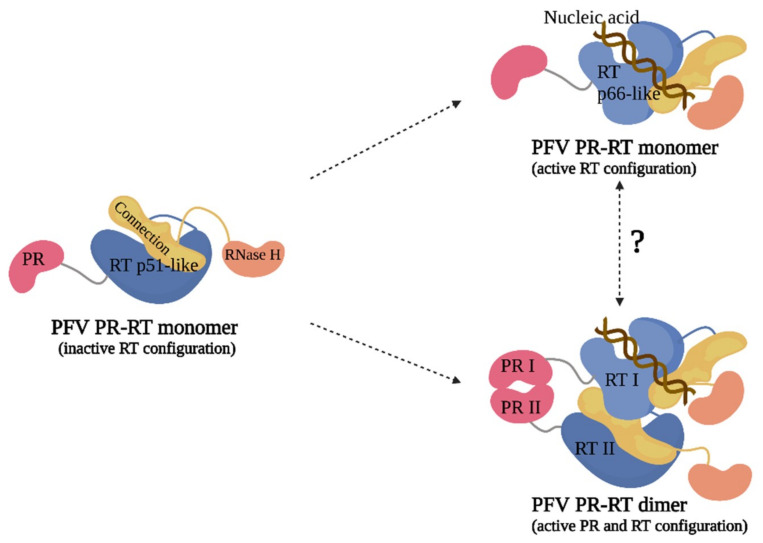
A model of the conformational rearrangement expected for the PFV RT and the dimerization of the PR.

**Table 1 viruses-13-01495-t001:** X-ray crystallographic data collection and refinement statistics.

	Selenomethionine (SeMet) PFV PR-RT (D24A+CSH) Mutant)	PFV PR-RT WT (D24A)
PDB ID	7KSE	7KSF
Wavelength (Å)	0.98	0.98
Resolution range (Å)	50.0–3.0 (3.18–3.0)	38.03–2.9 (3.004–2.9)
Space group	C2	C2
Unit cell (Å/deg)	240.38, 53.35, 74.92/90, 100, 90	228.21, 52.54, 75.43/90, 90.09, 90
Unique reflections	19100 (2990)	19443 (1752)
Multiplicity	30.4 (29.7)	5.1 (5.1)
Completeness (%)	99.7 (98.0)	96.1 (88.4)
Mean I/sigma(I)	10.6 (2.7)	7.1 (1.0)
Wilson B-factor (Å^2^)	70.3	92.1
R_merge_	0.331 (1.973)	0.128 (1.532)
R_meas_	0.337 (2.007)	0.157 (1.888)
R_pim_	0.061 (0.367)	0.064 (0.812)
CC_1/2_	0.996 (0.818)	0.993 (0.334)
R_work_	0.277 (0.382)	0.224 (0.370)
R_free_	0.312 (0.376)	0.258 (0.381)
Number of non-hydrogen atoms	5911	5915
macromolecules	5898	5905
ligands	1	1
solvent	12	9
Protein residues	743	743
RMS bonds (Å)	0.007	0.004
RMS angles (deg)	1.40	0.71
Ramachandran favored (%)	89.40	94.43
Ramachandran allowed (%)	10.05	5.30
Ramachandran outliers (%)	0.54	0.27
Rotamer outliers (%)	0.46	0.77
Clash score	23.68	11.65
Average B-factor (Å^2^)	85.66	103.78
macromolecules	85.71	103.78
ligands	110.53	123.65
solvent	62.54	103.14

Statistics for the highest-resolution shell are in parentheses. The SeMet dataset was merged from datasets collected from five separate crystals.

**Table 2 viruses-13-01495-t002:** Surface areas (Å^2^) buried at the interfaces of PFV PR-RT domains and subdomains. The extent of buried surfaces can provide an approximate estimate of the free energy of interaction, when the mixture of hydrophobic, hydrophilic, and electrostatic interactions is taken into account. Corresponding ΔG values (kcal/mol) estimated by jsPISA are given in parentheses.

	PR	PR-CTE	Fingers	Palm	Thumb	Connection	RNase H
PR	-	338 (−4.8)	632 (−7.5)	-	-	-	-
PR-CTE	338 (−4.8)	-	257 (1.0)	1009 (−17.4)	-	-	-
Fingers	632 (−7.5)	257 (1.0)	-	785 (−7.6)	-	212 (−1.7)	-
Palm	-	1009 (−17.4)	785 (−7.6)	-	359 (0.0)	547 (−7.2)	
Thumb	-	-	-	359 (0.0)	-	1216 (−17.9)	164 (−2.0)
Connection	-	-	212 (−1.7)	547 (−7.2)	1216 (−17.9)	-	425 (−3.3)
RNase H	-	-	-	-	164 (−2.0)	425 (−3.3)	-

## Data Availability

The data presented in this study are available in the Appendix A.

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
