# Peer review of "Crystal Structure of a Retroviral Polyprotein: Prototype Foamy Virus Protease-Reverse Transcriptase (PR-RT)"

_viruses, 2021, doi:10.3390/v13081495_

Round 1
Reviewer 1 Report
This paper reports the high-resolution crystal structure of a foamy virus PR-RT (without IN). There are oddities: the structure is of a monomer. Thus, the PR cannot be active. Furthermore, the RT region is folded in a conformation that is somewhat similar to the HIV-1 p51 subunit, and cannot bind nucleic acid in a catalytically useful way.
The structure is presented and discussed clearly, and compared in great detail with the HIV-1 and other retroviral PRs and RTs.
The authors need to clarify upfront that this is the structure of an inactive, monomeric, form – presumably the precursor, which needs to be extensively rearranged to form the mature active enzymes. This is laid out in the discussion (ll. 431-450) but is needed earlier too.
It would be important also to explain more about the enzymatic activities of the molecule being studied here – does it have good RT activity? RNase H activity? Protease activity? What is stated (ll. 132-139) deals with comparing the wt to the mutant version needed for expression, but how does it compare to a wt, fully active mature enzyme? If it is has good activity, do we then assume it is refolding to a “correct” structure with activity?
In other words, although the structure as presented is not consistent with RT or PR activity, can the molecule rearrange in solution to become active?
line 247: missing material
Author Response
Response to Viruses reviewer #1 comments
- The authors need to clarify upfront that this is the structure of an inactive, monomeric, form
– presumably the precursor, which needs to be extensively rearranged to form the mature
active enzymes. This is laid out in the discussion (ll. 431-450) but is needed earlier too.
Response from authors: We are grateful for your comments and apologize for not making this clearer in the original manuscript. In line 21, of the abstract, the statement, “in a configuration not competent for proteolytic or polymerase activity” has been added. Similar modifications have been added to line 37 and line 103.
- It would be important also to explain more about the enzymatic activities of the molecule
being studied here – does it have good RT activity? RNase H activity? Protease activity?
What is stated (ll. 132-139) deals with comparing the wt to the mutant version needed for
expression, but how does it compare to a wt, fully active mature enzyme? If it is has good
activity, do we then assume it is refolding to a “correct” structure with activity?
In other words, although the structure as presented is not consistent with RT or PR activity,
can the molecule rearrange in solution to become active?
Response from authors: We agree that these points should have been clearer in the original manuscript. In line 290, we have added the following paragraph: ‘The WT enzyme showed robust polymerase, processivity and ribonuclease activity (Figure S1B) that was consistent with data published in Boyer et al, 2004. However, because the PR-RT contains a mutation that inactivates PR, D24A, proteolytic activity was not determined’, to precede the comparison between the activities of the WT and the CSH mutant.
In line 246, the sentence is not finished.
- In Figure 3, all the structure drawings should contain the marks to indicate N- and C- termini.
- Figure 1 should contain the positions of the amino acid replacement in the CSH mutant.
- Figure 1 should also indicate the residues H507 and S584 as instability hotspots.
- Figure 2B is referred in line 174, while Figure 2A comes up first in line 235. The order of the figures should be arranged properly.
Response from authors: We agree with your comments. Appropriate modifications have been made that reflect the requested changes.
Reviewer 2 Report
The review result is attached.

Author Response
Response to Viruses reviewer #2 comments
- Line 35: The authors noted that the PFV RT is folded similar to the p51 subunit of HIV-1 RT with the Rnase H subdomain packed against the thumb and opposite the connection and palm subdomains. However, HIV-1 RT p51 structure in Figure 3B does not seem to contain the RNase H subdomain in orange. I could not get pints from this IMPORTANT section.
Response from authors: You are right about this. The RNase H domains of approximately half of the HIV-1 RT subunits are cleaved during maturation by HIV-1 PR. This is one of the steps needed to form the mature p66/p51 heterodimer. There is no structure of any full-length monomeric form of HIV-1 RT (either p66, or p66 folded to look like with p51 with an appended RNase H). Nor is there a structure of any form of dimeric RT in which RNase H is attached to the subunit folded into a p51-like configuration. Thus, neither the position nor the structure of the RNase H domain when it is attached to p51 is known. We therefore compared the folding of the polymerase domain of the PFV RT (fingers, palm, thumb, and connection) to the p51 of HIV-1 RT mindful of the fact that the RNase H domain is not present in the p51 subunit of the mature heterodimer (see lines 25-30).
- From line 146, the authors describe the structural characters found in the crystal structure of PFV PR-RT. Because the drawings lack the marks to identify the secondary structures shown in Figures 2 and 3, it is very difficult to follow the discussion in the document. Besides, the names for the sheets in Figure 2B are not consistent with those in the document; they are mentioned as b1PR and so on, but the structure drawing showed the names as b1 and as follows. Additionally, two short helices named αA’and αB’ are not marked on any structure drawings. In line 185, the new name b’3 comes up, which makes me further confused. The authors have to put the marks in the structure drawings consistent with those used in the document. This reviewer asks the authors to mark all the essential parts in structure drawings when they are noted in the document. The problems are not limited to Figures 2 and 3, but all the figures used in the manuscript have to be reconsidered.
Response from authors: Sorry for the mismatches and the confusion created. The numbering of the b sheets in PR have been modified and all other essential features of the structures have been annotated accordingly to make them clearer.
- In the section titled “Buried surface analysis of interactions” from 370, the authors discussed the strength of the interdomain interactions based on the sizes of the buried surfaces. However, the discussions are misleading. The size of the contact surface is not directly relating to the strength of the interaction. The authors have to compare the associated energies in the contact surfaces.
Response from authors: We agree with your comments that the relationship between the buried surface area is only an approximate measure and not a direct correlate of associated energy. We have therefore provided the corresponding DG values for these surfaces, estimated using program jsPISA (taking into account the nature of the interacting surfaces), to provide a context to the narrative.
- In the Discussion part, the authors should use the cartoons to make their findings summarized. Because of the lack of p66/p51 heterodimer structure, it is hard to get the points in the discussion in lines 442-456. Without a clear visualized summary of their findings, the manuscript hardly gives any significance to the readers who are not familiar with this particular type of virus protein. Overall, the manuscript gives the impression to give us just details of the structural characterization of PFV PRRT. Most of the readers may need to know the key insights into the structure and function of PFV PRRT, not the mere structure comparison with the relating proteins.
Response from authors: That is a great suggestion. A cartoon representation that summarizes the findings and puts them in a context has been added for clarity.
Round 2
Reviewer 1 Report
The authors have made edits that are responsive to my suggestions.
Reviewer 2 Report
The authors corrected and improved the manuscript according to the suggestions of the reviewer. The manuscript is now acceptable for publication.